# STAT1 and Its Crucial Role in the Control of Viral Infections

**DOI:** 10.3390/ijms23084095

**Published:** 2022-04-07

**Authors:** Manlio Tolomeo, Andrea Cavalli, Antonio Cascio

**Affiliations:** 1Department of Health Promotion Sciences, Maternal and Infant Care, Internal Medicine and Medical Specialties, University of Palermo, 90127 Palermo, Italy; antonio.cascio03@unipa.it; 2Computational and Chemical Biology, Italian Institute of Technology, 16152 Genova, Italy; andrea.cavalli@iit.it; 3Department of Pharmacy and Biotechnology, University of Bologna, 40126 Bologna, Italy

**Keywords:** STAT1, viral infection, immune response, Interferon

## Abstract

The signal transducer and activator of transcription (STAT) 1 protein plays a key role in the immune response against viruses and other pathogens by transducing, in the nucleus, the signal from type I, type II and type III IFNs. STAT1 activates the transcription of hundreds of genes, some of which have been well characterized for their antiviral properties. STAT1 gene deletion in mice and complete STAT1 deficiency in humans both cause rapid death from severe infections. STAT1 plays a key role in the immunoglobulin class-switch recombination through the upregulation of T-bet; it also plays a key role in the production of T-bet+ memory B cells that contribute to tissue-resident humoral memory by mounting an IgG response during re-infection. Considering the key role of STAT1 in the antiviral immune response, many viruses, including dangerous viruses such as Ebola and SARS-CoV-2, have developed different mechanisms to inhibit this transcription factor. The search for drugs capable of targeting the viral proteins implicated in both viral replication and IFN/STAT1 inhibition is important for the treatment of the most dangerous viral infections and for future viral pandemics, as shown by the clinical results obtained with Paxlovid in patients infected with SARS-CoV-2.

## 1. Introduction

Signal transducer and activator of transcription proteins (STATs) act both as signal transducers in the cytoplasm and as activators of transcription in the nucleus. Currently, seven types of STAT proteins are known; they are involved in different biological processes such as immunity, cell division, cell death and tumor formation [1,2,3].

STAT1 is the major mediator of the cellular response to interferons (IFNs) [4,5]. It plays a key role in the immune response against viruses and mycobacteria by transducing in the nucleus the signal from type I IFNs (IFNα, IFNβ, IFNε, IFNκ, and IFNω), type II IFN (IFNγ) and type III IFN (IFNλ). As with the other members of the STAT family, STAT1 is activated by tyrosine phosphorylation within the cytoplasm by a class of non-receptor tyrosine kinases called Janus kinases (JAKs and TYK2) associated with IFN receptors [6,7,8].

Upon IFN stimulation, phosphorylated STAT1 molecules dimerize through reciprocal phosphotyrosine (pTyr)-SH2 interactions. STAT1 dimers are transferred in the nucleus where they activate gene transcription [9,10].

The specific role of STAT1 in the IFN signaling pathways was clearly demonstrated by Meraz et al. in 1996, in mice deficient in STAT1 [11]. These mice exhibited a complete state of insensitivity to IFNα or IFNγ and were unable to defend themselves against different types of infections. The study revealed an unexpected level of specificity for STAT1 in effecting cellular responses to IFNs. Similar results were obtained by Durbin et al. in 1996 [12].

## 2. Structure of STAT1

In 1998, Chen et al. determined the crystal structure of the DNA complex of a STAT1 homodimer at 2.9 Å resolution [13]. As with the other STAT proteins, STAT1 consists of six different regions: (1) a helical N-terminal domain (ND) which is important for interactions between adjacent STAT dimers on DNA; (2) a coiled-coil (CC) domain that interacts with regulatory proteins; (3) a DNA-binding domain (DBD) for the recognition of specific DNA-sequences of target genes; (4) a helical linker (LK) domain involved in nuclear export and DNA binding; (5) an Src homology 2 (SH2) domain for receptor binding and dimerization; (6) a C-terminal transactivation domain (TAD) that contains specific residues that are phosphorylated upon transcriptional activation. There are two isoforms of STAT1 produced by alternative splicing of its transcript: STAT1α (91 kDa) and STAT1β (84 kDa) (Figure 1) [14]. STAT1α is constituted by 750 amino acids and represents the main form of STAT1. STAT1β is a short form and is constituted by 712 amino acids. STAT1α has two phosphorylation sites at residues tyrosine 701 (Y701) and serine 727 (S727) located at the TAD, whereas STAT1β lacks part of the TAD and contains only the Y701 phosphorylation site.

The β-isoform was initially believed to be transcriptionally inactive. However, studies in gene-modified mice have demonstrated that STAT1β can induce gene expression, although it is less efficient than STAT1α in mediating the IFNγ-dependent innate immune defense [15]. Unphosphorylated STAT1 was initially reported to be monomeric [16]. Further studies showed that unphosphorylated STAT1 can be homo- or hetero-dimer [17]. In fact, STAT1 was reported to coimmunoprecipitate with STAT2 and STAT3 without cytokine stimulation [18].

To explore the nature of unphosphorylated STATs, Mao et al. crystallized an unphosphorylated human STAT1 [19]. They observed that STAT1 can exist in a dimeric form prior to its activation, and the ND interactions contribute to the dimer stability. The connecting region between the ND and the core fragment (constituted by CC, DB, LK and SH2 domains) is flexible and allows two interconvertible orientations of the core fragments, termed “antiparallel” or “parallel,” as determined by the SH2 domain orientations [19]. Mutations in either the CC/DB domain interface or the ND dimer interface block dimerization of unphosphorylated molecules [20].

## 3. Activation of STAT1

STAT1 is critical in signal transduction from the type I IFNs and the type II IFN. It is also activated in response to type III IFN and to several interleukins and growth factors [21]. STAT1 activation requires the phosphorylation of Y701 and S727 as well as the methylation of amino-terminal arginine. The inhibition of STAT1 arginine methylation or the mutation of STAT1 Arg-31 result in a prolonged half-life of STAT1 tyrosine phosphorylation.

Interferons bind and signal through transmembrane receptors (IFNRs) that activate JAK1/2 and TYK2, which, in turn, phosphorylate STAT1 leading to its dimerization/activation. As reported in Figure 1, type I, type II and type III IFNs bind to specific receptors. JAK1 and TYK2 are activated by type I and type III IFNs. This leads to the recruitment and phosphorylation of STAT1 and STAT2; they associate to form a heterodimer, which in turn recruits IRF9 to form the so-called interferon-stimulated gamma factor-3 (ISGF3). Type II IFN activates JAK1 and JAK2, which recruit and phosphorylate STAT1. Phosphorylated STAT1 homodimers form the gamma interferon activated sequence (GAF). ISGF3 and GAF are transferred into the nucleus and bind to specific DNA consensus sequences (the interferon-stimulated response element, or ISRE, and the gamma interferon activated sequence GAS) sited in the promoter and the enhancing sequences of IFN-stimulated genes (ISGs), leading to the activation of their transcription [22,23,24] (Figure 2). IFNα is produced prevalently by plasmacytoid dendritic cells (pDC), whereas IFNβ expression can be induced in almost all cells in the body in response to microbial products. IFNγ is produced mainly by T helper lymphocytes, macrophages and natural killer (NK) cells [25].

There is some degree of tissue specificity in the production of type I, type II and type III IFNs. In the brain, IFNα/β is readily produced after infection with various RNA viruses, whereas expression of IFNλ is low in this organ. The IFNλ response is prominent in the stomach, intestine and lungs. In the liver, virus infection induces the expression of IFNα/β, IFNλ genes and IFNγ; however, the contribution of spleen cell systemic IFNγ production is greater as an absolute number than in the liver [26,27].

These IFN/JAK/STAT signaling pathways are considered the canonical pathways. In addition, non-canonical IFN pathways have been described. Non-canonical IFN type I signaling pathways are similarly activated by IFNs binding to the extracellular regions of the dimeric IFNαR1 and IFNαR2 complex, leading to JAK1/TYK2 activation, but they diverge from that point, specifically, by not involving STAT activation by the JAKs. The main non-canonical IFN pathways identified are the mitogen-activated protein kinase (MAPK) and the phosphoinositide 3-kinases (PI3K)/mammalian target of rapamycin (mTOR) pathways. MAPK signaling pathways are composed of three subfamilies, including ERK, SAPK/JNK and p38 MAPK. MAPK and PI3K/mTOR pathways have been shown to elicit effects on ISG transcription and mRNA translation while also having some interaction with STATs in the canonical cascade [28]. These non-classical IFN-induced effectors play critical roles in ISG transcription, independent of or in conjunction with the canonical pathways, eliciting specific biological responses.

## 4. STAT1 Nuclear Import and Export

Following IFN stimulation, STAT1 is transferred from the cytoplasm into the nucleus. This transfer is mediated by a complex constituted by two proteins: importin-α and importin-β (Figure 3).

This complex is essential for nuclear STAT1 import. STAT1 binds importins in the cytosol, and the resulting trimeric complex moves through the nuclear pores [29]. Specific nuclear localization signals (NLSs) located within the DBD of STAT1 and constituted by arginine/lysine-rich elements, one in each STAT monomer, mediate the direct interaction of STAT dimers with importin-α, which activates the nuclear import process. Residue K413, L407 and the N-terminal domain of STAT1 are essential for this interaction, suggesting an important role for STAT1 spatial conformation in nuclear translocation [30]. The binding of IFN to the cell surface receptor causes the recruitment and phosphorylation of STAT1. Through spontaneous dissociation and reassociation, the activated STAT1 molecules constantly oscillate between the parallel and antiparallel dimer conformations (Figure 2). NLSs are in an inactive state in the unphosphorylated STAT1 monomer. After binding to NLS on the STAT1 dimer, importin-α binds to importin-β in the cytoplasm. This complex (STAT1/importin-α/importin-β) interacts with the nuclear pore, and the movement that drives the import substrate complex into the nucleus appears to be generated between importin-β and structures of the nuclear pore [31] (Figure 3). It has been observed that STAT1 dimers are unable to enter the nucleus under specific conditions (e.g., stimulation with angiotensin II [32], TNF-α [33] or HGF [34]). This suggests that in some conditions, STAT1 may serve specific cytoplasmic functions [35]. For example, in the cytoplasm, STAT1 interacts with Runx2, inhibiting the nuclear localization of this transcription factor that is essential for osteoblast differentiation [36].

STAT1 dephosphorylation on tyrosine 701 is required for rapid nuclear export. Dephosphorylation is induced in the nucleus by tyrosine phosphatase TC45 [37]. After the interaction with the DNA and the activation of transcription, STAT1 conformation shifts from parallel to antiparallel dimers; this conformation is susceptible to dephosphorylation by the nuclear phosphatase TC45. Nuclear protein export is mediated by the nuclear protein exportin1 that interacts with a region of STAT1 that is located between residues 392 to 413 and is known as the nuclear export sequence (NES) [38]. Exportin 1 shows homology to importin-β and can specifically interact with both the NES motif and the Ran (RAs-related nuclear protein) GTPase. Ran is a small, 25 kDa protein that is involved in transport out of the cell nucleus during interphase. It is a member of the Ras superfamily, and its interaction with exportin1 is required for STAT1 nuclear export [39].

## 5. Components of the STAT1 Transcriptional Complex

After its interaction with the DNA, STAT1 needs other components to induce transcription. These co-factors are: (1) CBP/p300; (2) Nmi; (3) MCM-5 and (4) BRCA-1.

### 5.1. CBP/p300

CBP/p300 is composed of two transcriptional co-activating proteins: p300 and CBP. Both these proteins interact with numerous transcription factors and act to increase the expression of their target genes. These co-activating proteins increase gene expression in three ways: (i) by relaxing the chromatin structure; (ii) by recruiting the basal transcriptional complex, including RNA polymerase II, to the promoter; and (iii) by acting as adaptor molecules. CBP/p300 can bind to both the C-terminal and the N-terminal regions of STAT1, enhancing its transcription activity. The interaction between the carboxyl terminus of STAT1α and the E1A-binding domain of CBP/p300 is required for the transcriptional activation of STAT1α after stimulation with IFNγ [40].

### 5.2. Nmi

N-myc and STAT interactor (Nmi) is a protein that indirectly influences specific transcriptional events. Nmi augments STAT1-mediated transcription in response to inflammatory cytokines. Moreover, Nmi enhances the association of CBP/p300 with STAT1. Therefore, Nmi can potentiate STAT1-dependent transcription and augment the recruitment of other coactivator proteins [41].

### 5.3. MCM5

Minichromosome maintenance 5 (MCM5) is a nuclear protein recruited by STAT1 for transcription activation. MCM5 is a helicase with an ATPase activity involved in DNA replication; it interacts with the TAD region of STAT1, enhancing the transcription induced by STAT1. The interaction between STAT1 and MCM5 requires two specific residues in MCM5: R732 and K734 [42]. Mutations in these residues do not allow interaction between STAT1 and MCM5 in response to IFNγ [43].

### 5.4. BRCA1

Breast cancer gene 1 (BRCA1) tumor suppressor acts in concert with STAT1 to differentially activate the transcription of a subset of IFNγ target genes and mediates growth inhibition by this cytokine. This interaction plays an important role in the IFNγ-dependent tumor surveillance system. After treatment with IFNγ, induction of p21WAF1, a cyclin-dependent kinase inhibitor, is synergistically activated by STAT1 and BRCA1. The mechanism of this transcriptional synergy involves interaction between BRCA1 and TAD, including Ser-727 [44].

## 6. STAT1 Gene Targets

Activated STAT1 leads to the transcription of hundreds of ISGs. These genes are characterized by two specific DNA consensus sequences, GAS and ISRE, sited in the promoter and enhancer regions. The GAS sequence consists of a palindromic sequence ‘TTC’ and ‘GAA’, separated by three nucleotides (5′-TTCNNN/GAA-3′) that interact with GAF, which is composed of a phosphorylated STAT1 homodimer. The ISRE sequence is composed of repeats of the motif 5′-TTTC-3′ or its complement 5′-GAAA-3′ separated by one or two nucleotides [45]. STAT1 transcriptional activation of ISGs leads to a remarkable antiviral state by producing proteins that interfere with different phases of the viral life cycle. The antiviral mechanisms of the proteins produced by some antiviral ISGs have been well characterized, but for most ISGs little is known about the antiviral mechanism of their product. Different viruses are often targeted by unique sets of ISGs. Furthermore, each viral species can induce the activation of multiple antiviral genes with different inhibitory activities [46]. The best-studied ISGs endowed with antiviral effects include Viperin, Mx1, Mx2, IFITM, CH25H, TRIM proteins, BST-2 and CXCL9.

### 6.1. Viperin

Virus inhibitory protein, endoplasmic-reticulum-associated, IFN-inducible (Viperin or RSAD2) is one of the most important antiviral effectors activated by STAT1 [47]. Viperin has been reported to inhibit a broad spectrum of DNA and RNA viruses, including herpesviruses (HCMV), flaviviruses (HCV, West Nile virus (WNV) and Dengue virus), an alphavirus (Sindbis virus), an orthomyxovirus (Influenza A virus), a paramyxovirus (Sendai virus), a rhabdovirus (Vesicular Stomatitis Virus) and a retrovirus (HIV-1) [48]. Viperin normally resides in the cytosolic face of the endoplasmic reticulum (ER) and in lipid droplets (LDs), both of which often serve as platforms for viral replication complexes. Viperin catalyzes the conversion of cytidine triphosphate (CTP) to 3′-deoxy-3′, 4′-didehydro-cytidine triphosphate (ddhCTP), which serves as a chain terminator during virus replication. Viperin can also promote viral protein degradation and interfere with the Golgi-dependent trafficking of soluble proteins and promote the release of immature capsids [49]. Viperin inhibits virus release from the plasma membrane of infected cells by reducing plasma membrane fluidity. This alteration is due to the intracellular interaction of Viperin with farnesyl diphosphate synthase (FPPS), an enzyme essential for isoprenoid biosynthesis [50,51]. The reduced activity of FPPS alters the fluidity of the membrane, thus interfering with the budding of the virus. Inhibition of HIV-1 particle release is believed to occur by a similar mechanism [51].

### 6.2. Mx1 and Mx2

The product of the murine myxovirus resistance (Mx) gene was one of the first antiviral proteins described. There are two different Mx proteins in human cells, MxA (Mx1 in mice) and MxB (Mx2 in mice), that are endowed with different antiviral mechanisms. MxA belongs to the class of dynamin-like, large guanosine-5’-triphosphatases that are involved in intracellular vesicle trafficking. MxA is active against orthomyxoviruses, paramyxoviruses, vesicular stomatitis virus and hepatitis B virus (HBV) [52]. MxA binds and mislocates the proteins of viral nucleocapsids into membrane-associated, large perinuclear complexes. The interactions between oligomeric MxA/nucleocapsid complexes stimulate GTPase activity, which directs them to sites of degradation [53]. It has been observed that the expression of MxA mRNAs after treatment with IFNα was higher in control cells than in STAT1 knockout cells, indicating that STAT1 is involved in the early induction of this ISG in response to IFNα [54]. MxB protein inhibits vesicular stomatitis virus and hantavirus, but not influenza virus [52]. Moreover, MxB is necessary for the complete antiviral activity of IFNα against HIV-1 and HIV-2. In fact, the MxB protein blocks the nuclear entry of the HIV-1 reverse-transcribed genome, therefore ultimately inhibiting its integration into the DNA of infected cells [55]. IFNλ induces STAT1-dependent MxA and MxB expression, but MxB showed the highest increase after stimulation with IFNλ [56].

### 6.3. IFITM Proteins

STAT1 is a transcription factor for IFN-induced transmembrane (IFITM) proteins that can inhibit the endocytic-fusion events of several viruses. In humans, the IFITM family of proteins is composed of four members: IFITM1, IFITM2, IFITM3 and IFITM5. IFITM1 inhibits SARS-coronavirus (CoV), Hepatitis C Virus (HCV) and the Ebola and Marburg filoviruses [57,58]. IFITM2 and IFITM3 inhibit HIV-1 virus production and virus entry, while IFITM1 inhibits only virus production [59]. IFITM1, IFITM2 and IFITM3 play an important role in SARS-CoV-2 defense. However, IFITM proteins also appear to be important for SARS-CoV-2 entry into human lung cells. Therefore, the degree of severity of SARS-CoV-2 infection may be a result of these opposing activities [60]. Recent studies suggest that IFITM proteins can alter cell surface membrane rigidity to prevent the fusion of the viral and cellular membranes [61].

### 6.4. CH25H

The cholesterol-25-hydroxylase (CH25H) gene is upregulated by both type I and type II IFNs via STAT1 [62,63]. The protein product is an enzyme that converts cholesterol into 25-hydroxycholesterol (25HC). The antiviral effects of 25HC occur at the step of virus–host membrane fusion. Moreover, its antiviral activity may also result, at least in part, from its involvement in the regulation of the sterol biosynthesis pathway. 25HC inhibits sterol biosynthesis in both an autocrine and a paracrine manner [64]. The sterol biosynthesis pathway generates isoprenoids that play a critical role for protein prenylation, which is important for the life cycle of several viruses. For example, the hepatitis delta virus large antigen is modified by prenylation, and preventing this modification abolishes infectious particle production [65]. Prenylation is also required for HCV replication [66]. The antiviral effects of 25HC on the herpes simplex virus 1 (HSV-1) and cytomegalovirus (MCMV) have also been described [62].

### 6.5. TRIM Proteins

The tripartite motif (TRIM) family of proteins is composed of more than 70 members that exhibit a wide range of activities. Members of the TRIM protein family display antiviral properties, and their expression depends on the activation of JAK-STAT1/2 signaling [67]. TRIM19, also known as promyelocytic leukemia protein (PML), inhibits the replication of many unrelated viruses such as poliovirus, influenza virus, rabies virus, adeno-associated virus (AAV) and HIV. TRIM19 forms specific structures with other proteins, such as Sp100 and Daxx, that produce important antiviral effects. For example, TRIM19 and Daxx are able to inhibit the reverse transcription of HIV-1 [68]. TRIM5α has been shown to be responsible for the resistance of primate cells to diverse retrovirus infections. TRIM5α is another member of the TRIM family able to induce the premature dissociation of the capsid of HIV-1 and inhibit reverse transcription [69]. TRIM22 has been shown to inhibit various viruses. Its ubiquitination can inhibit encephalomyocarditis virus replication. TRIM22 can inhibit HBV replication by interfering with the HBV core promoter, influenza A virus by degradation of the influenza A virus nucleoprotein, and human HCV by the degradation of its NS5A protein [70].

### 6.6. BST-2

Bone marrow stromal antigen 2 (BST2) (also known as tetherin) is another potent interferon-inducible host antiviral factor that blocks viral replication by inhibiting enveloped virus budding from the surface of infected cells. BST2 inhibits virus budding by using two membrane anchors to trap virions on the plasma membrane. BST2 is effective against many enveloped viruses [71].

### 6.7. CXCL9

Chemokine (C-X-C motif) ligand 9 (CXCL9) is a monokine induced by IFNγ/STAT1. It plays a role in inducing chemotaxis, in the differentiation and multiplication of leukocytes, and in the recruitment of immune cells such as cytotoxic lymphocytes, natural killer (NK) cells and macrophages [72].

## 7. Role of STAT1 in the Immunoglobulin Class-Switch Recombination

B lymphocytes can undergo class-switch recombination to produce a single, specific antibody (immunoglobulin—Ig) isotype, including IgM, IgA, IgE or one of the IgG subclasses. IgMs are the first antibodies produced during a new infection. They increase for several weeks and then decline as IgG production begins. IgGs represent the basis of long-term protection against microorganisms to prevent re-infection.

Cytokines such as IL-4, IFNγ and transforming growth factor β (TGF-β) play a key role in B cell differentiation by directing the isotype specificity of class-switch recombination. IFNγ and IL-4 act as reciprocal regulatory agents in determining Ig isotype responses. In mice, the IFNγ/STAT1 pathway induces the expression of antibodies of the IgG2a isotype and inhibits the production of IgG3, IgG1, IgG2b and IgE. In contrast, IL-4 has powerful effects in promoting switching to the expression of IgG1 and IgE by activating transcription factors such as STAT6, but it also markedly inhibits IgM, IgG3, IgG2a and IgG2b [73]. TGF-β appears to selectively stimulate class-switch recombination to IgG2b.

IgG is the major and most efficient isotype produced during antiviral immune responses [74], and T-bet (T-box expressed in T cells) has been shown to be critical for this process [75]. T-bet is a member of the T-brain1 subfamily of T-box transcriptional factor genes that plays an important role in Th1 immune response and in the repression of Th2 and Th17 programs [76]. T-bet controls the differentiation and maturation of CD8+ T cells [77] and, in B cells, it is required for IgG class switching [75]. The expression of T-bet in B cells is driven by the engagement of their B-cell receptor (BCR), IFNγR, toll-like receptor (TLR) 7 or CD40 and STAT1 activation. Specifically, viral proteins bind to the BCRs of follicular B cells while viral RNA binds to TLR7. Viral infections also induce the production of IFNγ by different cell types that bind with IFNγR of B cells. These stimuli induce the activation of STAT1 that upregulates T-bet, which induces the class switching of B cells toward IgG2a production in mice [78] and IgG1/IgG3 in humans [79]. T-bet promoter is a direct STAT1 target, and the stimulation of B cells with IFNγ causes an increase in T-bet expression of more than 60-fold [80]. In humans, T-bet+ B cells are a normal component of the human immune system that is overrepresented during viral infections and vaccination with virus vaccines. Furthermore, T-bet+ B cells persist indefinitely in mice and humans, indicating their role as memory B cells. In fact, T-bet+ B cells express several surface markers specific to memory B cells in mice [81]. Moreover, they show antibody diversification, which is typical of memory cells, and they are important in the switch to IgG. Several data suggest the possibility that T-bet− memory B cells function to monitor the lymphatic system, while T-bet+ counterparts perform surveillance of tissues supplied by blood and may contribute to tissue-resident humoral memory by mounting an IgG response during re-infection [82]. Depletion and knockout experiments in mouse models show that T-bet+ B cells are required for optimal immune responses to bacteria, parasites and viruses, including influenza virus, cytomegalovirus, vacciniavirus and Friend virus infections [83].

## 8. Human STAT1 Deficiency

Inborn errors of human STAT1 with loss-of-function (LOF) mutations cause three types of primary immunodeficiencies: (1) autosomal recessive (AR), characterized by biallelic complete LOF mutations of STAT1 and complete STAT1 deficiency, (2) AR partial STAT1 deficiency caused by biallelic hypomorphic STAT1 mutations, and (3) autosomal dominant (AD) heterozygous STAT1 deficiency.

The AR characterized by biallelic complete LOF mutations of STAT1 and complete STAT1 deficiency is a rare autosomal recessive primary immunodeficiency. Patients affected by this form of STAT1 mutation have a complete block of the IFN/STAT1 pathway and are affected by lethal intracellular bacterial and viral diseases. In addition, analysis of natural killer (NK) cells from these patients shows a loss of response to IFNα and reduced NK cytolytic activity, suggesting that the STAT1 pathway may be important for the function of these cells [84]. Ex vivo experiments have shown that there is no response to IFNγ in terms of IL-12 production and HLA class II induction responses in fresh blood leukocytes cultured in vitro, and IFNα/β did not suppress HSV and vesicular stomatitis virus replication in fibroblasts. Furthermore, the production of IL-12, IFNγ and TNF-α are also reduced in these patients [85]. These data are in keeping with the observation that STAT-1 increases the levels of both IL-12 and TNF-α in mice [86,87].

The recessive homozygous form of partial STAT1 deficiency is characterized by impaired IFNγ and IFNα/β signaling. Patients suffer from severe but treatable intracellular bacterial and viral infections. In addition, this recessive STAT1 deficiency is characterized by altered IL-27 and IFNλ signaling pathways, which are conditions that may contribute to susceptibility to infections [88].

Heterozygous dominant-negative mutations of STAT1 are responsible for autosomal-dominant Mendelian susceptibility to mycobacterial diseases and chronic mucocutaneous candidiasis. Mutations can be localized to the DB domain, the SH2 domain and the CCD domain. These heterozygous mutations have a dominant-negative effect on GAS transcriptional activity following IFNγ stimulation [89,90].

## 9. STAT1 in Viral Infection and Viral Mechanisms of Immune Evasion

The production of IFNs and the activation of STAT1 represent key steps in the immune response against viruses. IFNα is produced prevalently by pDCs that are a subtype of dendritic cells that links innate and adaptive immunity. After activation, pDCs migrate from tissues to lymphoid organs, where they prime naive T cells to initiate acquired immune responses. pDC are mainly activated upon the triggering of endosomal TLR7 and TLR9 that recognize motifs that are conserved between large classes of microbial pathogens. TLRs activate two different pathways: (i) the nuclear factor-κB (NF-κB) pathway, through the recruitment of the adaptor MyD88 that leads to the production of inflammatory cytokines such as IL-1β, IL-6, TNF-α and IL-12, and (ii) the IFN regulatory factors (IRFs) pathway that leads to type I IFN production. Three family members of the IRF family, IRF3, IRF5 and IRF7, are critical in producing type I IFNs downstream of TLRs. A fourth family member, IRF9, regulates interferon-driven gene expression [91]. Type I IFNs, in turn, activate STAT1, which induces the transcription of ISGs [92].

The IFNγ and IFNα/β signal pathways cross-talk at multiple levels, and IL-12 plays a relevant role in this interconnection. IL-12 is produced by several populations of cells, including DCs, monocytes, macrophages, neutrophils and B cells. IL-12 is composed of two protein chains, p35 and the p40, that are encoded by different genes. The transcription of IL-12p35 mRNA is regulated by STAT1, as shown by the reduction in IL-12p70 but not IL-12p40 secretion in bone marrow DCs from STAT1−/− mice [86]. The main role of IL-12p70 is to stimulate naive CD4+ T cells to differentiate into T helper type 1 (Th1) cells that are required for host defense against intracellular viral and bacterial pathogens. To induce this differentiation, IL-12 binds to the IL-12 receptor (IL-12R) and promotes the phosphorylation, homodimerization and nuclear translocation of STAT4 that induces the production of IFNγ. This IFN drives Th1 differentiation and increases IL-12p70 mRNA transcription through the activation of STAT1. Gene mutations related to this pathway have been associated with increased susceptibility to viral infections [93,94] and other infections such as mycobacterial and salmonellae [95,96,97].

IFNγ plays a crucial role in the production and activation of microbicidal molecules in macrophages. Microbicidal ability activated by IFNγ includes the induction of the NADPH-dependent phagocyte oxidase (NADPH oxidase) system, priming for nitric oxide (NO) production, and upregulation of anti-microbial lysosomal enzymes [98]. Macrophages kill viruses and intracellular pathogens primarily by the production of reactive oxygen species (ROS) and reactive nitrogen intermediates (RNI) via the IFNγ/STAT1 pathway that induces the production of the NADPH oxidase system and inducible nitric oxide synthase (iNOS) [99]. Indeed, IFNγ induces the transcription of the gp91phox and p67phox subunits of the NADPH oxidase complex, which upregulates ROS production in phagocytes. Interestingly, the pharmacologic inhibition of the JAK/STAT1 signaling pathway blocks ROS production and the induction of genes encoding the NADPH oxidase components [100]. Considering the crucial role of IFNs and STAT1 in antiviral defense, viruses have developed different mechanisms to block IFN pathways (Table 1).

### 9.1. Degradation of STAT1

One of the most important mechanisms to block IFN pathways is the proteosome-mediated degradation of STAT1. For example, the paramyxoviruses simian virus 5 (SV5) and human parainfluenza virus type 2 block IFN signaling in human cells by specifically targeting STAT1 for proteasome-mediated degradation. The expression of the SV5 and human parainfluenza virus type 2 structural protein V induces the degradation of STAT1, indicating that this protein plays the central role in evading antiviral defense [101].

This is also a mechanism used by the mumps virus (MuV) to block IFN pathways, as demonstrated by the ability of this virus to reduce constitutive STAT1 levels 10 h after infection. STAT1 levels were restored when MuV-infected cells were treated with a proteosome inhibitor [102]. Similar to SV5, HPIV2 and MuV, the Newcastle disease virus (NDV) encodes a V protein that induces the degradation of STAT1 [103].

The herpes simplex viruses 1 (HSV-1) and 2 (HSV-2) are the human herpesviruses that cause recurrent cold sores and genital herpes. They are double-stranded DNA viruses that establish lifelong infections in hosts and alternate between two programs of gene expression: productive replication and latent infection. HSV-2 can suppress the production of ISGs through its protein ICP22 (infected cell protein 22). This protein promotes the ubiquitination and degradation of STAT1, STAT2 and IRF9, resulting in the blockage of ISGF3 nuclear translocation [104]. HSV-1 encodes a protein called Infected Cell Protein 0 or ICP0, that directly interfaces with component proteins of the ubiquitin pathway to inactivate host immune defenses. ICP0 functions as an activator in the balance between HSV-1 latency and reactivation: when ICP0 is synthesized, the protein overcomes STAT1-dependent cellular repression of HSV-1 such that the virus can complete its replication cycle; on the contrary, when ICP0 is not synthesized, STAT1-dependent cellular repression silences the HSV-1 genome [105].

### 9.2. Phosphorylation Inhibition of STAT1

Many viruses are able to prevent the phosphorylation of STAT1. For example, human metapneumovirus (hMPV) can cause persistent lung infections evading host viral clearance by inhibiting the type I interferon response through the inhibition of STAT1 phosphorylation [107].

Proteins of the Nipah virus (NiV) and the Hendra Paramyxovirus-family virus can inhibit the phosphorylation and nuclear translocation of STAT1 and STAT2. P, V and W proteins encoded by NiV can bind to STAT1 through their amino-terminal domain, inhibiting its activation and nuclear translocation [108].

The measles virus, a paramyxovirus of the Morbillivirus genus, causes an acute childhood illness that infects over 40 million people and leads to the deaths of more than 1 million people annually. Moreover, this virus can induce an important immune suppression in infected people. The measles V protein prevents the IFNα/β and IFNγ -induced transcriptional response by reducing STAT1 nuclear accumulation through its V protein [125] and by inhibiting the phosphorylation of STAT1 through its P protein [109].

The vaccinia virus encodes for a protein (VH1) that can block the IFN/STAT1 pathway by inhibiting STAT1 phosphorylation/activation and inducing its dephosphorylation and nuclear translocation [110]. The protein NS5 of the mosquito-borne Japanese Encephalitis flavivirus also induces the dephosphorylation of STAT1, preventing STAT1 nuclear translocation and the transcription of its gene targets [111].

The human cytomegalovirus (HCMV) has developed mechanisms that can counteract the control of infection by inhibiting the phosphorylation of tyrosine 701 of STAT1. This process is due to the induction by HCMV infection of SHP2 (Src homology region 2 domain-containing phosphatase), a cellular ubiquitous phosphatase involved in the regulation of IFNγ-mediated tyrosine phosphorylation that directly dephosphorylates nuclear STAT1 [112].

The non-structural protein 5A (NS5A) is a hydrophilic zinc-binding and proline-rich phosphoprotein important for RNA replication and HCV assembly [126]. This protein is derived from an HCV polyprotein that undergoes post-translational processes by a viral protease and plays an important role in HCV resistance to IFNα by interacting with and suppressing the phosphorylation of STAT1 in infected hepatocyte cells [113].

Rotavirus (RV) replicates efficiently in intestinal epithelial cells in vivo despite the activation of a local host IFN response. RV replication and spread in intestinal epithelial cells occur despite exogenous stimulation of the STAT1-mediated IFN signaling pathway. This depends on the ability of RV to inhibit IFN-mediated STAT1 tyrosine 701 phosphorylation in human intestinal epithelial cells by way of NSP1, a non-structural RNA-binding protein that is a component of early replication intermediates [114].

The human Coronavirus OC43 (HCoV-OC43) infection can specifically reduce STAT1 and STAT3 phosphorylation. In fact, when Vero cells were infected with HCoV-OC43, STAT1 and STAT3 phosphorylation decreased sharply within 48 h. However, the mechanism underlying the dephosphorylation of STAT1 and STAT3 in response to HCoV-OC43 infection is not yet known [115].

### 9.3. Inhibition of STAT1 Nuclear Import/Export and STAT1 Binding to DNA

The Rabies virus RNA polymerase complex consists of the large protein (L) and its cofactor, phosphoprotein (P). STAT1 can interact with its N-terminal portion with the C-terminal domain of P. This phosphoprotein can block STAT1 nuclear translocation and STAT1 binding to the DNA promoter of IFN-responsive genes [118,119].

The filoviruses, Ebola virus (EBOV) and Marburg virus (MARV), cause frequently lethal viral hemorrhagic fever. These viruses encode for two proteins that efficiently inhibit IFNα/β pathways: VP35 and VP24. VP35 inhibits the transcriptional activation of IRF-3. In EBOV, VP24 alters the nuclear accumulation of activated STAT1 by interacting with importin (karyopherin) alpha1 and blocking its interaction with STAT1 [120]. Of interest, the MARV VP24 protein does not detectably block STAT1 nuclear import; unlike EBOV, MARV infection inhibits STAT1 and STAT2 phosphorylation. Thus, despite their similarities, there are fundamental differences in the methods by which these deadly viruses counteract the IFN system [116].

Chikungunya virus (CHIKV) is a mosquito-borne alphavirus associated with large outbreaks on the African, Asian, European, and American continents. In most patients, infection results in high fever, rash and chronic arthralgia. CHIKV effectively inhibits the IFN/STAT1 pathway with its non-structural protein 2 (nsP2). In particular, the C-terminal domain of nuclear nsP2 specifically inhibits the IFN response by promoting the nuclear export of STAT1. In the nucleus, nsP2 seems to compete with STAT dimers for binding to the ISRE and GAS DNA elements [123].

### 9.4. Inhibition of STAT1 Methylation

Pegylated IFN has been used for the treatment of chronic hepatitis C in combination with Ribavirin and for the treatment of chronic hepatitis B. However, many patients have not shown a good response to this treatment. This could be related to the ability of HCV virus to block the IFN pathway by increasing the levels of protein phosphatase 2A (PP2A) in hepatocytes. PP2A is increased in chronic hepatitis B by the HBV protein HBVX. PP2A inhibits STAT1 by blocking the enzyme arginine methyltransferase 1 (PRMT1), which catalyzes STAT1 methylation. Hypomethylated STAT1 is less active because it is bound by its inhibitor, PIAS1 [124].

### 9.5. Multi-Step Inhibition of IFN Pathway

Severe acute respiratory syndrome coronavirus (SARS-CoV) causes a severe acute respiratory syndrome that is often fatal. SARS-CoV has proteins that can overcome the immune response by blocking the IFN/STAT1 pathway. For example, the N protein of SARS-CoV inhibits the synthesis of interferon, while ORF 3b and ORF 6 proteins inhibit both interferon synthesis and signaling. Furthermore, the ORF 6 protein is able to block the phosphorylation/activation of STAT1 and its nuclear translocation [127] by binding and disrupting the STAT1 import complex. [121].

Enterovirus (EV) infections are responsible for mild respiratory illness, hand, foot, and mouth disease (HFMD), acute hemorrhagic conjunctivitis, aseptic meningitis, myocarditis, severe neonatal sepsis-like disease and the acute flaccid paralysis epidemic. Among them, enterovirus A71 (EV-A71) is a major pathogen that causes hand, foot, and mouth disease, which can be fatal with neurological complications in children. Several EV-71 non-structural proteins are able to block the IFN/STAT1 pathway. For example, EV-A71 2A protease (2A^pro^) inhibits IFN-γ-induced serine phosphorylation of STAT1, while EV-A71 3D^pro^ attenuation of IFN-γ signaling was accompanied by a STAT1 decrease in mouse embryonic fibroblasts transfected with 3D^pro^ [117]. Among EV-A71 viral proteins, 2B is involved in the degradation of importin (karyopherin) alpha1 blocking the translocation of STAT1 into the nucleus upon IFN-α stimulation [122].

## 10. STAT1 and SARS-CoV-2 Infection

The severe acute respiratory syndrome coronavirus 2 (SARS-CoV-2)-associated coronavirus disease 2019 (COVID-19) pandemic has been the subject of several studies showing that this virus can affect IFN/STAT1 signaling. SARS-CoV-2 infects the respiratory tract, resulting in pneumonia in most cases and acute respiratory distress syndrome (ARDS) in about 15% of cases [128]. Two large studies have shown that IFN signaling is impaired in patients with severe SARS-CoV-2 infection. Bastard et al. reported that at least 101 of 987 patients with life-threatening COVID-19 pneumonia had neutralizing immunoglobulin G (IgG) autoantibodies (auto-Abs) against IFNω, against IFNα or against both at the onset of critical disease. These auto-Abs were not found in 663 individuals with asymptomatic or mild SARS-CoV-2 infection and were present in only 4 of 1227 healthy individuals [129].

Zhang et al. observed that 23 of 659 patients with life-threatening COVID-19 pneumonia had known or new genetic defects at eight loci in the TLR3- and IRF7 genes. These genotypes were silent until infection with SARS-CoV-2 that caused life-threatening pneumonia [130].

In 2020, Wu et al. found that SARS-CoV-2 infection can decrease the mRNA levels of IFN-stimulated downstream cytokines as well as the IFN-triggered phosphorylation level of STAT1. These effects seem to be correlated to 3CLpro (Mpro), a critical proteolytic enzyme for the virus life cycle. 3CLpro activity yields non-structural proteins such as RNA-dependent RNA polymerase, helicase and ribonucleases that are crucial for genome replication and coronavirus virion production by cleaving two large polyproteins called 1a and 1ab. Co-immunoprecipitation analysis showed that SARS-CoV-2 3CLpro interacts with STAT1 by inducing its autophagic degradation. Of interest, the autophagy/autolysosome inhibitors 3-methyladenine and bafilomycin A1 were able to restore the levels of STAT1 [106]. In particular, 3CLpro could promote the colocalization between STAT1 and microtubule-associated protein 1 light chain 3 beta (LC3B), a protein that plays a central role in the autophagy pathway, where it functions in autophagy substrate selection and autophagosome biogenesis (Figure 4). In addition, the overexpression of 3CLpro reduced the ubiquitination/activation of retinoic acid-inducible gene I (RIG-I), which is an important viral RNA sensor of cells. Together, these data may suggest that the treatment of COVID-19 with IFN could be effective, at least at the earliest stage of SARS-CoV-2 infection. In contrast, a meta-analysis of experimental studies carried out by Wijaya et al. showed that the use of JAK inhibitors can reduce the risk of mortality in hospitalized patients with COVID-19, inducing a clinical improvement [131]. In fact, mortality in patients with COVID-19 has been linked to excessive production of proinflammatory cytokines, a condition called “cytokine storm”, that leads to ARDS aggravation, tissue damage and multi-organ failure. The production of type I IFN, together with other factors such as nuclear factor kB (NFkB) and activation protein 1 (AP-1) transcription factor, induce the expression of genes encoding inflammatory cytokines and chemokines that in SARS-CoV-2 infection can be the cause of a “cytokine storm” [132]. In conclusion, both the substitution of interferons and the blocking of interferon signaling through JAK-STAT inhibition to limit cytokine storms have been proposed for the treatment of COVID-19.

Recently, Rincon-Arevalo et al., by studying the expression of phosphorylated and unphosphorylated STAT1 in patients with mild or severe COVID-19, suggested a model to understand when to prefer a treatment with interferons or with drugs capable of blocking interferon signaling [133]. They observed that both Siglec-1 (a well described downstream molecule in interferon signaling) and STAT1 levels in plasmablasts and monocytes were higher in patients mildly affected by COVID-19 than in those with severe infection. Similar results were obtained by Doehn et al., who found an increased expression of Siglec-1 in monocytes from patients with mild SARS-CoV-2 infection compared to patients with severe disease [134]. The authors concluded that patients with low Siglec-1 expression and an inability to increase their antiviral response (severe disease) may benefit from IFN treatment, while patients with higher Siglec-1 expression and risk of cytokine storm may benefit from treatment with JAK/STAT inhibitors.

The crucial role of 3CLpro in SARS-CoV-2 replication and STAT1 degradation makes this protease an attractive target for novel therapies. Paxlovid (Nirmatrelvir/ritonavir) is a new drug developed by Pfizer for the oral treatment of COVID-19. Nirmatrelvir (PF-07321332) blocks the activity of 3CLpro by binding directly to the catalytic cysteine (Cys145) residue of the cysteine protease enzyme, while Ritonavir serves to slow down the metabolism of Nirmatrelvir with cytochrome enzymes to maintain higher circulating concentrations of the main drug [135]. Therefore, unlike other anti-SARS-CoV-2 drugs (such as Molnupiravir and Remdesivir), Nirmatrelvir not only blocks SARS-CoV-2 replication but may also reverse the immunosuppression caused by the effects of 3CLpro on IFN and STAT1.

On 5 November 2021, Pfizer announced the results of a scheduled interim analysis of a study on Paxlovid in patients with SARS-CoV-2. The therapeutic efficacy of Paxlovid was markedly superior to that of other drugs used for the treatment of COVID-19, including Molnupinavir, with an 89% reduction in risk of hospitalization or all-cause mortality compared to a placebo. Among patients treated within the first three days of symptoms, by day 28 of follow-up, 3/389 were hospitalized with no deaths in the Paxlovid group, compared to 27/385 hospitalized with seven deaths in the placebo group (“Pfizer’s Novel COVID-19 Oral Antiviral Treatment Candidate Reduced Risk Of Hospitalization Or Death By 89% In Interim Analysis Of Phase 2/3 EPIC-HR Study”. Pfizer Inc., 5 November 2021).

## 11. Conclusions

Several studies have shown the central role that STAT1 plays in the immune response against viral infections by transducing in the nucleus the signal from type I, type II and type III IFNs. STAT1 activates the transcription of genes with crucial antiviral properties, and the selective gene deletion of STAT1 in mice or the presence of loss-of-function mutations of STAT1 in humans both cause rapid death from severe infections. Furthermore, over the course of coevolution, many viruses have become able to efficiently block the IFN-STAT1 pathway by the proteosome-mediated degradation of STAT1 or by blocking the activation process of STAT1 or its nuclear localization. This may have contributed to the diffusion of severe and fatal epidemics such as Ebola virus, SARS-CoV, and more recently the SARS-CoV-2 pandemic. The clinical results obtained with Paxlovid for the treatment of the SARS-CoV-2 infection (89% reduction in the risk of COVID-19-related hospitalization with no deaths) compared with a placebo or other drugs used against COVID-19, might suggest that targeting viral proteins, such as 3CLpro, that are implicated in both viral replication and in IFN/STAT1 inhibition could be an important tool for the treatment of the most dangerous viral infections and for future viral pandemics. The two-fold mechanism of action of such drugs, i.e., antiviral activity and the ability to prevent STAT1 degradation, may represent their key hallmark and pave the way towards innovative antiviral strategies.

## Figures and Tables

**Figure 1 ijms-23-04095-f001:**
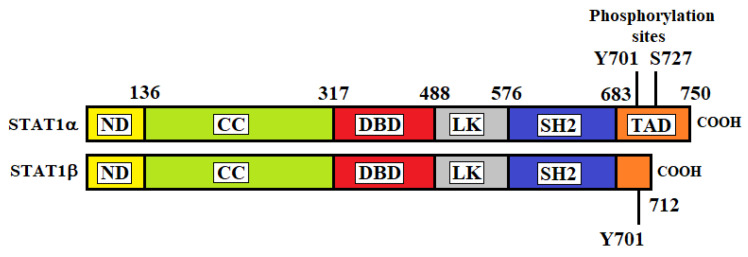
Structures of STAT1α and STAT1β. ND, N-terminal domain; CC, coiled-coil domain; DBD, DNA binding domain; LK, linker domain; SH2, Src homology-2 domain; TAD, transactivation domain.

**Figure 2 ijms-23-04095-f002:**
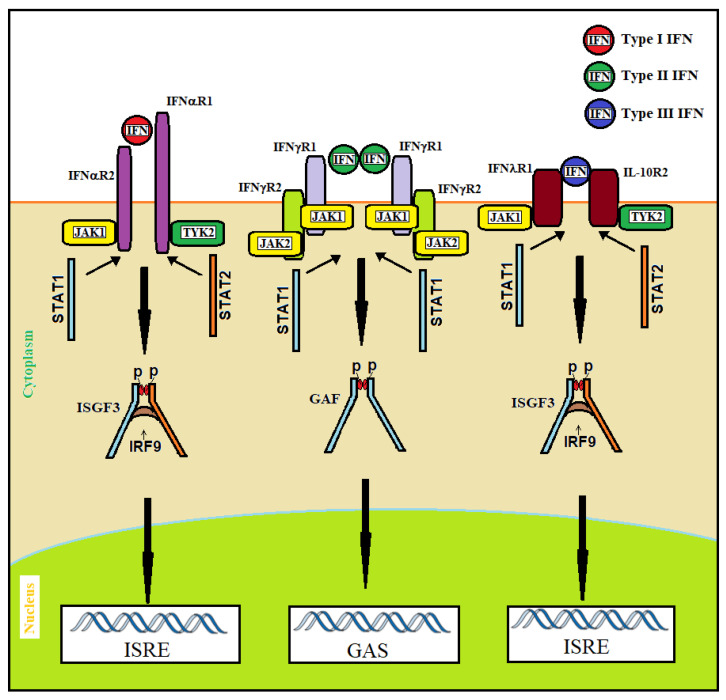
IFN-signaling pathways. Type I IFNs act through IFNαR1 and IFNαR2 heterodimers, type II IFN through dimers of heterodimers constituted by IFNγR1 and IFNγR2, and type III IFN through IL-10R2 and IFNλR1 heterodimers. After binding with their receptors, type I and type III IFNs trigger phosphorylation of JAK1 and TYK2, which in turn phosphorylate the receptors at specific intracellular tyrosine residues. This leads to the recruitment and phosphorylation of STAT1 and STAT2 that associate to form a heterodimer, which in turn recruits IRF9 to form the so-called interferon-stimulated gamma factor-3 (ISGF3). In addition to signaling through STAT1–STAT2 heterodimers, type I IFN can signal through STAT1 homodimers. The binding of type II IFN to IFNγR1 and IFNγR2 leads to the dimerization of the complex and the phosphorylation of JAK1 and JAK2. The transphosphorylation of IFNγR1 and IFNγR2 by JAK1 and JAK2 causes the recruitment and phosphorylation of STAT1. Phosphorylated STAT1 homodimers form the gamma interferon activated sequence (GAF). Both ISGF3 and GAF translocate to the nucleus and bind to specific DNA consensus sequences (GAS for GAF and ISRE for ISGF3) sited in the promoter and the enhancing sequences of IFN-stimulated genes with antiviral activity.

**Figure 3 ijms-23-04095-f003:**
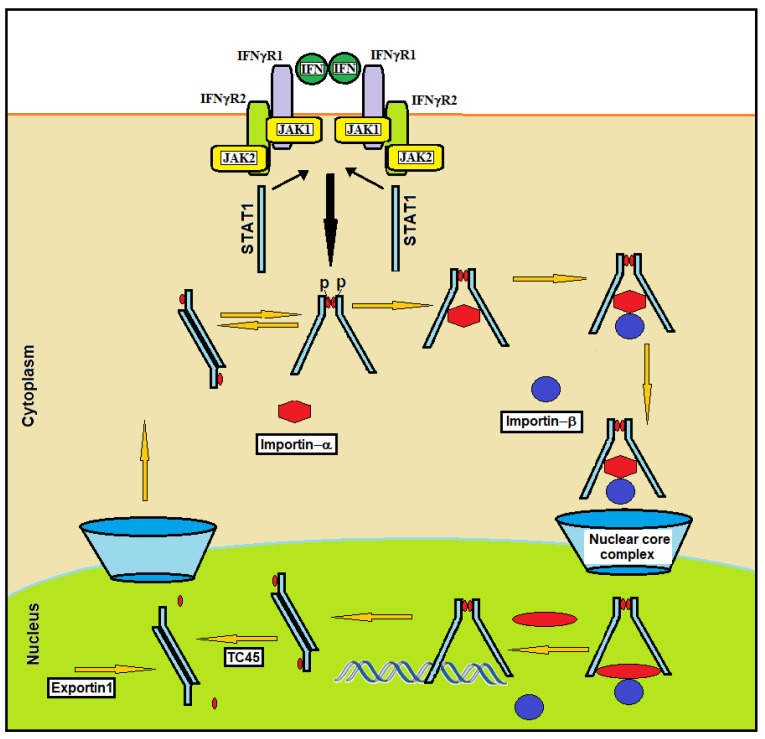
Model of STAT1 import and export after IFNγ stimulation. The binding of IFN to the cell surface receptor causes the recruitment and phosphorylation of STAT1. Through spontaneous dissociation and reassociation, the activated STAT1 molecules constantly oscillate between the parallel and antiparallel dimer conformations. The importin-α recognizes and binds to the NLS on the STAT1 dimer. After binding to the STAT1 dimer, importin-α binds to importin-β in the cytoplasm. This complex (STAT1/importin-α/importin-β) interacts with the nuclear pore, and the movement that drives the import substrate complex into the nucleus appears to be generated between importin-β and structures of the nuclear pore. After interaction with the DNA and activation of transcription, STAT1 conformation shifts from parallel to antiparallel dimers, which is a conformation susceptible to dephosphorylation by the nuclear phosphatase TC45. Unphosphorylated STAT1 is then exported to the cytoplasm by exportin1.

**Figure 4 ijms-23-04095-f004:**
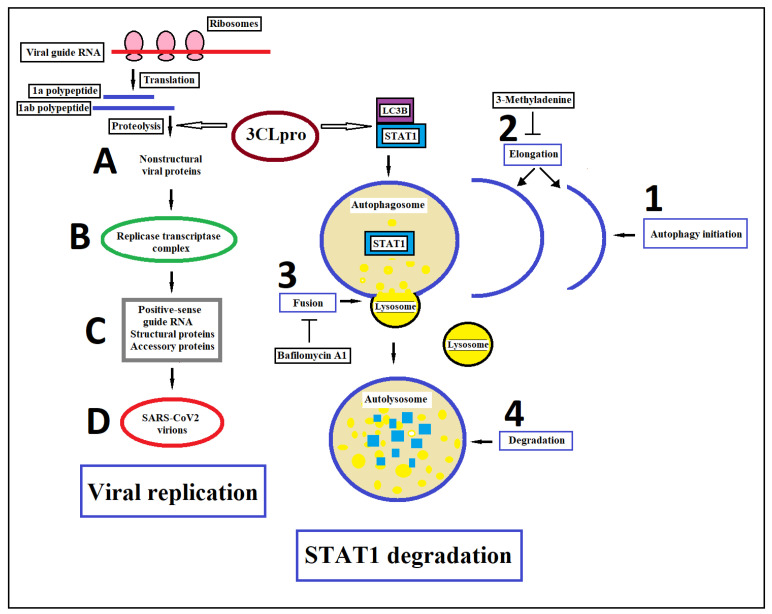
Schematic representation of 3CLpro effects in viral replication and STAT1 degradation. SARS-CoV-2 is a positive-sense single-stranded RNA virus. Upon fusion of the viral and host cell membranes, viral genomic RNA is released in the cytoplasm. (A) the viral RNA initiates the translation of co-terminal polypeptides 1a and 1ab. These polyproteins are subsequently cleaved by 3CLpro protease into non-structural proteins such as RNA-dependent RNA polymerase, helicase and ribonucleases; (B) the coronavirus replicase-transcriptase complex is an assembly of viral and cellular proteins that mediates the synthesis of genome- and subgenome-sized mRNAs in the virus-infected cell; (C) in the course of the SARS-CoV-2 infective cycle, the replicase-transcriptase complex amplifies the genomic RNA and synthesizes mRNAs that encode the structural proteins of the virus and proteins that are non-essential for replication in cell culture but appear to confer a selective advantage in vivo (accessory proteins); (D) production of new virions. The autophagic process is divided into four distinct stages: (1) Initiation, (2) autophagosomal formation (elongation), (3) autophagosome-lysosome fusion, and (4) autophagolysosome formation and cargo degradation. LC3B is a central protein in the autophagy pathway, where it functions in autophagy substrate selection and autophagosome biogenesis. 3CLpro could promote the colocalization between STAT1 and LC3B causing the autophagic degradation of STAT1, as shown by the rescue of STAT1 by autophagy/autolysosome inhibitor 3-methyladenine and bafilomycin A1.

**Table 1 ijms-23-04095-t001:** Mechanisms by which viruses block the STAT1 pathway.

Mechanisms	Virus	Proteins Involved in STAT1 Inhibition	References
Degradation of STAT1	Human parainfuenza virus 2	V protein °	[101]
	Simian virus 5	V protein °	[101]
	Mumps virus	V protein °	[102]
	Newcastle disease virus	V protein °	[103]
	Herpes viruses	ICP22 °, ICP0 °	[104,105]
	SARS-CoV-2	3CLpro °	[106]
Phosphorylation inhibition and dephosphorylation of STAT1	Human metapneumovirus	-	[107]
	Nipah virus	P, V and W proteins °	[108]
	Measles virus	P protein °	[109]
	Vaccinia virus	VH1 °	[110]
	Encephalitis flavivirus	NS5 °	[111]
	Cytomegalovirus	SHP2 *	[112]
	HCV	NS5A °	[113]
	Rotaviruses	NSP1 °	[114]
	HCoV-OC43	-	[115]
	Marbung virus	-	[116]
	Enteroviruses	2A^pro^ °	[117]
Inhibition of STAT1 Nuclear import and STAT1 binding to DNA	Rabies virus	P protein °	[118,119]
	Ebola virus	VP24 °	[120]
	SARSV-CoV	ORF6 °	[121]
	Enteroviruses	2B °	[122]
Inhibition of STAT1 Nuclear export	Chikungunya virus	nsP2 °	[123]
Inhibition of STAT1 Methylation	HCV and HBV	PP2A *	[124]

° viral proteins; * cellular proteins activated by the virus; - the protein involved in STAT1 inhibition is not known.

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
