# Peer review of "STAT1 and Its Crucial Role in the Control of Viral Infections"

_ijms, 2022, doi:10.3390/ijms23084095_

Round 1
Reviewer 1 Report
The manuscript of Tolomeo et al., entitled: ‘STAT1 and its crucial role in the control of viral infections’, gives a comprehensive overview of the role of signal transducer and activator transcription (STAT) 1 in the control of viruses. The structure, activation, translocation to the nucleus, co-factors, gene targets, role in humoral response and viral immune evasion are the topics that are extensively described. Overall, this manuscript is well written and discusses the complex regulatory system of the innate and adaptive immune responses to defend against viral infections. However, some remarks need to be addressed.
Title: The authors mention ‘viral infections’ in the title. However, it would be of great interest to the reader if more viruses are used as an example to explain the activation and the antiviral potential of the STAT1 pathway. At this moment, this manuscript relevant examples are limited.
Introduction: Clear overview of the different proteins involved in the IFN signaling pathway and their function.
Activation of STAT1: The authors give a structured overview of the role of STAT1 in the signal transduction of type I, II and III interferon (IFN). It would be of interest to the readers to include tissue specificity and outline in time of the different types of IFN. Is it known in literature if STAT1 is also tissue/organ specific? It is also worth mentioning that in addition to the classical JAK-STAT1 signaling pathway to activate type I IFN, other signaling factors have a role in IFN activation, including isoforms of protein kinase C, MAPK, PI3K and the ERK/MAPK pathway. Under which conditions is classical and the non-classical pathways activated?
STAT1 nuclear import and export:
Lines 150-152: Does importin-a interact with importin-b prior to the interactions of importin-b with proteins of the nucleopore complex? If so, please rearrange order of the sentences.
Line 153: The authors mention a potential cytoplasmic function of STAT1 under specific conditions. Is there some information available in literature whether and when these conditions also might occur in vivo?
Figure 2 is a standard figure of the IFN signaling pathways. It would be novel and more of interest to the readers to replace this figure or add another figure to explain what happens during the nuclear import/export and in the nucleus.
Components of the STAT1 transcriptional complex: This paragraph gives a clear overview of the co-factors involved in the induction of DNA transcription; Is it know if the type of IFN have an effect on which co-factor becomes activated?
STAT1 gene targets: Please add examples of viruses and references to explain the antiviral potential of each protein. E.g. line 217, some viruses use the ER and LD for viral replication, which virus family?
Human STAT1 deficiency:
Line 348: which subtypes of ‘fresh blood cells’ were cultured?
Line 349: change ‘coltured’ into ‘cultured’
STAT1 in viral infection and viral mechanisms of immune evasion:
Lines 366-369: Move to ‘Activation of STAT1’ paragraph.
Line 370: How do DCs become activated during viral infections?
Table 1: Mumps virus is a Paramyxovirus. How did the authors decide to include which virus? It seems that these viruses are chosen ad random. What about Herpes simplex viruses? Enteroviruses? Rearrange this table, add relevant examples covering different types of virus families.
STAT1 and SARS-CoV2 infection: What is known of other Coronaviruses, such as MERS, SARS1 or OC43 ? It would be of interest to compare SARS-CoV2 with other coronaviruses, to demonstrate differences and similarities.
What is known about the mRNA vaccines of SARS-CoV2 and the induction of the STAT1 pathway?
Author Response
To the Editor of
International Journal of Molecular Sciences (IJMS).
Dear Editor,
I am sending you a copy of my revised manuscript entitled: “STAT1 and its Crucial Role in the Control of Viral Infections.” to be considered for publication in the section "Molecular Microbiology " of International Journal of Molecular Sciences (IJMS). The manuscript has been revised according to the referees' comments and I explain, point by point, the details of the revisions to the manuscript and my responses to the referees’ comments.
In the revised manuscript, changes made according to the reviewers' suggestions are marked in yellow. Table 1 has been modified according to the suggestions of the reviewers and a new figure (Figure 2) has been added as requested by the reviewer 1. The first two chapters have been shortened according to the request of the reviewer 3. However, in Chapter 2, although shortened, I had to add information requested by reviewer 1.
REVIEWER 1
- Title: The authors mention ‘viral infections’ in the title. However, it would be of great interest to the reader if more viruses are used as an example to explain the activation and the antiviral potential of the STAT1 pathway. At this moment, this manuscript relevant examples are limited.
In the revised manuscript, we have added a number of viruses (Herpes viruses, Enteroviruses, Rotaviruses, HCoV-OC43, Merbung virus, chikungunya virus and others) in the Chapter “STAT1 in viral infection and viral mechanisms of immune evasion” and in the Chapter “STAT1 gene targets”.
- Activation of STAT1: The authors give a structured overview of the role of STAT1 in the signal transduction of type I, II and III interferon (IFN). It would be of interest to the readers to include tissue specificity and outline in time of the different types of IFN. Is it known in literature if STAT1 is also tissue/organ specific? It is also worth mentioning that in addition to the classical JAK-STAT1 signaling pathway to activate type I IFN, other signaling factors have a role in IFN activation, including isoforms of protein kinase C, MAPK, PI3K and the ERK/MAPK pathway. Under which conditions is classical and the non-classical pathways activated?
We included information on the tissue specificity of type I, II, and III IFNs at the end of Chapter 3, and in the same chapter we included information on non-classical IFN pathways.
- Lines 150-152: Does importin-a interact with importin-b prior to the interactions of importin-b with proteins of the nucleopore complex? If so, please rearrange order of the sentences.
The following sentence has been added in chapter 4: “After binding to NLS on the STAT1 dimer, importin-α binds to importin-β in the cyto-plasm. This complex (STAT1/importin-α/importin-β) interacts with the nuclear pore and the movement which drives the import substrate complex into the nucleus appears to be generated between importin-β and structures of the nuclear pore”.
- The authors mention a potential cytoplasmic function of STAT1 under specific conditions. Is there some information available in literature whether and when these conditions also might occur in vivo?
An excessive increased of bone mass was observed in STAT1-deficient mice. This increase is caused by an excessive osteoblast differentiation. Actually, STAT1 interacts with Runx2 inhibiting the nuclear localization of Runx2, which is an essential transcription factor for osteoblast differentiation. This function of STAT1 does not require the Tyr 701 that is phosphorylated when STAT1 becomes a transcriptional activator. This is an example in which STAT1 acts in the cytoplasm attenuating the activity of another transcription factor causing specific alterations in vivo in STAT1-deficient mice . We added this information in the revised manuscript.
- Figure 2 is a standard figure of the IFN signaling pathways. It would be novel and more of interest to the readers to replace this figure or add another figure to explain what happens during the nuclear import/export and in the nucleus.
We modified Fig 2 and added a figure that describes the import and export process of STAT1.
- Components of the STAT1 transcriptional complex: This paragraph gives a clear overview of the co-factors involved in the induction of DNA transcription; Is it know if the type of IFN have an effect on which co-factor becomes activated?
As reported in the "References" the main STAT1 co-factors were studied after IFNgamma stimulation.
- STAT1 gene targets: Please add examples of viruses and references to explain the antiviral potential of each protein. E.g. line 217, some viruses use the ER and LD for viral replication, which virus family?
We have added more virus examples and related references.
- Line 348: which subtypes of ‘fresh blood cells’ were cultured?
Patient’s blood leukocytes. We added this cell type in the text.
- Line 349: change ‘coltured’ into ‘cultured’
We changed “coltured” in “cultured”
- Lines 366-369: Move to ‘Activation of STAT1’ paragraph.
We have moved these lines in the Activation of STAT1’ paragraph.
- Line 370: How do DCs become activated during viral infections?
In the revised manuscript we have added the following sentence: “pDC are mainly activated upon triggering of endosomal TLR7 and TLR9 that recognize motifs that are conserved between large classes of microbial pathogens”.
- Table 1: Mumps virus is a Paramyxovirus. How did the authors decide to include which virus? It seems that these viruses are chosen ad random. What about Herpes simplex viruses? Enteroviruses? Rearrange this table, add relevant examples covering different types of virus families.
Table 1 was compiled on the basis of the mechanism by which viruses block the STAT1 pathway. Thus, viruses were included not on the basis of virus family types but on the type of mechanism by which STAT1 is inhibited. As observed in the table, viruses from different families can have the same mechanism of action on STAT1 and the same virus can have two or more mechanisms of STAT1 inhibition. Furthermore, in the same family of viruses only a few can be endowed with STAT1 inhibition mechanisms .However, as suggested by reviewer 1, we added more viruses in Table 1 including herpes simplex virus and enterovirus.
- STAT1 and SARS-CoV2 infection: What is known of other Coronaviruses, such as MERS, SARS1 or OC43 ? It would be of interest to compare SARS-CoV2 with other coronaviruses, to demonstrate differences and similarities.
The effects of Sars-CoV(1) on STAT1 are reported in the manuscript. Regarding MERS-CoV no direct effects on STAT1 have been described, although accessory proteins 3, 4a, 4b, and 5 of MERS have shown the ability to inhibit the type I interferon response but not STAT1. In contrast, HCoV-OC43 infection can specifically reduce STAT1 and STAT3 phosphorylation. In fact, when Vero cells were infected with HCoV-OC43 STAT1 and STAT3 phosphorylation decreased sharply by 48 h. We added this information in the revised manuscript. However, the mechanism underlying the dephosphorylation of STAT1 and STAT3 in response to HCoV-OC43 infection is not yet known.
- What is known about the mRNA vaccines of SARS-CoV2 and the induction of the STAT1 pathway?
No data are known on the effects of SARS-CoV-2 RNA vaccines on STAT1 pathway.
Reviewer 2 Report
The manuscript is very well written and easy to read. The authors have done a great job
My only comment: I think the authors could make a summary table of Section 6.
Author Response
To the Editor of
International Journal of Molecular Sciences (IJMS).
Dear Editor,
I am sending you a copy of my revised manuscript entitled: “STAT1 and its Crucial Role in the Control of Viral Infections.” to be considered for publication in the section "Molecular Microbiology " of International Journal of Molecular Sciences (IJMS). The manuscript has been revised according to the referees' comments and I explain, point by point, the details of the revisions to the manuscript and my responses to the referees’ comments.
In the revised manuscript, changes made according to the reviewers' suggestions are marked in yellow. Table 1 has been modified according to the suggestions of the reviewers and a new figure (Figure 2) has been added as requested by the reviewer 1. The first two chapters have been shortened according to the request of the reviewer 3. However, in Chapter 2, although shortened, I had to add information requested by reviewer 1.
Reviewer 3 Report
The paper by Tolomeo and colleagues is a review covering many aspects of the protein STAT1 and its function in human phisyology.
There is a very long introductive part (chapter 1-5) which could be probably rendered more synthetic, also considering that most of the considerations exposed are
well established since many years (indeed the supporting references are often at least 20 years old), and could be skipped or summarized quickly.
Chapter 6 is instead much more informative and updated, while chapter 7 is probably to long, even because its topic is slightly less on topic with respect to the rest of the
paper. Chapter 9 again is more interesting and focused, as well as chapter 10, even if not all the referred articles deal directly with STAT1, and the distinction between effects of SARS-COV-2 on STAT1 and effects of pre-existing conditions on IFN pathway should be expressed in a much clearer way.
On the other hand, please consider that, looking at the abstract, the readers may expect that
the review is mainly focused at these two chapter, especially 10, and may be a little disappointing at finding them only at the very end of the long review. I think that it would be more
polite to decrease the emphasis given to Sars-Cov-2 in the abstract.
some minor points:
table 1 completely lacks references, please add.
line 513:'about 15% of cases': please add a reference
Author Response
To the Editor of
International Journal of Molecular Sciences (IJMS).
Dear Editor,
I am sending you a copy of my revised manuscript entitled: “STAT1 and its Crucial Role in the Control of Viral Infections.” to be considered for publication in the section "Molecular Microbiology " of International Journal of Molecular Sciences (IJMS). The manuscript has been revised according to the referees' comments and I explain, point by point, the details of the revisions to the manuscript and my responses to the referees’ comments.
In the revised manuscript, changes made according to the reviewers' suggestions are marked in yellow. Table 1 has been modified according to the suggestions of the reviewers and a new figure (Figure 2) has been added as requested by the reviewer 1. The first two chapters have been shortened according to the request of the reviewer 3. However, in Chapter 2, although shortened, I had to add information requested by reviewer 1.
REVIEWER 3
- There is a very long introductive part (chapter 1-5) which could be probably rendered more synthetic, also considering that most of the considerations exposed are well established since many years (indeed the supporting references are often at least 20 years old), and could be skipped or summarized quickly.
In the revised manuscript, we have reduced Chapter 1 "Introduction". In addition, we have reduced Chapter 3 "Activation of STAT1" considering that a lot of information reported in this Chapter was also reported in Figure 2.
- Chapter 6 is instead much more informative and updated, while chapter 7 is probably to long, even because its topic is slightly less on topic with respect to the rest of the paper. Chapter 9 again is more interesting and focused, as well as chapter 10, even if not all the referred articles deal directly with STAT1, and the distinction between effects of SARS-COV-2 on STAT1 and effects of pre-existing conditions on IFN pathway should be expressed in a much clearer way.
In Chapter 7 we explain the central role of STAT1 in immunoglobulin switching and in production of T-bet- memory B cells, thus the role of STAT1 in immune memory. This is extremely important because it explains how immune memory is impaired in infections caused by viruses that can inhibit STAT1 (such as SARS-CoV-2) and can explain the frequent relapse of Covid-19 in the same subject.
.
- On the other hand, please consider that, looking at the abstract, the readers may expect that the review is mainly focused at these two chapter, especially 10, and may be a little disappointing at finding them only at the very end of the long review. I think that it would be more polite to decrease the emphasis given to Sars-Cov-2 in the abstract.
In the revised abstract we reduced the emphasis given to Sars-Cov-2.
- table 1 completely lacks references, please add.
We added references in table 1.
- line 513:'about 15% of cases': please add a reference
We added a reference after the sentence “'about 15% of cases”.
We confirm that neither the manuscript nor any parts of its content are currently under consideration or published in another journal.
All authors have approved the manuscript and agree with its submission to IJMS.
Best Regards
Manlio Tolomeo
Department of Health Promotion Sciences, Maternal and Infant Care, Internal Medicine and Medical Specialties, University of Palermo, Italy. Via del Vespro 129, 90127, Palermo, Italy, email: manlio.tolomeo@policlinico.pa.it; Telephone: (+39) 091 6554002 - (+39) 3288149354.
Round 2
Reviewer 3 Report
The paper has been modified and improved in response to my considerations and in this version can be accepted for publication